# XY-Tokenizer: Mitigating the Semantic-Acoustic Conflict in Low-Bitrate Speech Codecs

## Abstract

Speech codecs serve as bridges between speech signals and large language models. An ideal codec for speech language models should not only preserve acoustic information but also capture rich semantic information. However, existing speech codecs struggle to balance high-quality audio reconstruction with ease of modeling by language models. In this study, we analyze the limitations of previous codecs in balancing semantic richness and acoustic fidelity. We propose **XY-Tokenizer**, a novel codec that mitigates the conflict between semantic and acoustic capabilities through multi-stage, multi-task learning. Experimental results demonstrate that XY-Tokenizer achieves performance in both semantic and acoustic tasks comparable to that of state-of-the-art codecs operating at similar bitrates, even though those existing codecs typically excel in only one aspect. Specifically, XY-Tokenizer achieves strong text alignment, surpassing distillation-based semantic modeling methods such as SpeechTokenizer and Mimi, while maintaining a speaker similarity score of 0.85 between reconstructed and original audio. The reconstruction performance of **XY-Tokenizer** is comparable to that of BigCodec, the current state-of-the-art among acoustic-only codecs, which achieves a speaker similarity score of 0.84 at a similar bitrate.

## 1 Introduction

In recent years, large language models (LLMs) (Achiam et al., 2023; Yang et al., 2024a) have achieved significant advancements in natural language processing, showcasing remarkable capabilities in understanding and generating text for fluent and natural conversations. Consequently, speech large language models (Speech LLMs) have garnered increasing attention (Zhang et al., 2023a; Chu et al., 2024; Défossez et al., 2024). A critical component of Speech LLMs is the speech codec, which transforms continuous speech signals into discrete tokens, aligning with the token-based approach of LLMs (Zeghidour et al., 2021; Défossez et al., 2022; Kumar et al., 2023; Zhang et al., 2023b; Défossez et al., 2024). Acoustic codecs trained through residual vector quantization GAN (RVQ-GAN) capture the details of the audio waveform and allow for high-quality synthesis (Défossez et al., 2022; Kumar et al., 2023; Wang et al., 2023; Xin et al., 2024). Self-supervised learning (SSL) models (Devlin et al., 2019) trained with masked language modeling (MLM) capture contextual dependencies in speech, making them widely used in Speech LLMs (Hsu et al., 2021; Chung et al., 2021; Chen et al., 2022; Chiu et al., 2022). Additionally, automatic speech recognition (ASR) models trained on large-scale supervised datasets align well with the text modality (Radford et al., 2023), and their discrete representations are often utilized as inputs for Speech LLMs (Zeng et al., 2024; Ding et al., 2025).

Semantic tokens, typically derived from discretized self-supervised learning (SSL) models, are considered to exhibit high alignment with text while leading to poor reconstruction. In contrast, acoustic tokens often derived from speech codecs trained through residual vector quantization GAN (RVQ-GAN), are recognized for capturing the details of the audio waveform, enabling high-quality synthesis, but they do not demonstrate strong alignment with text (Borsos et al., 2023). An ideal speech codec should effectively model both semantic and acoustic information. SpeechTokenizer employs semantic distillation, utilizing the output of the first layer of residual vector quantization (RVQ) to distill representations from a teacher SSL model (Zhang et al., 2023b). Similarly, the Mimi codec in

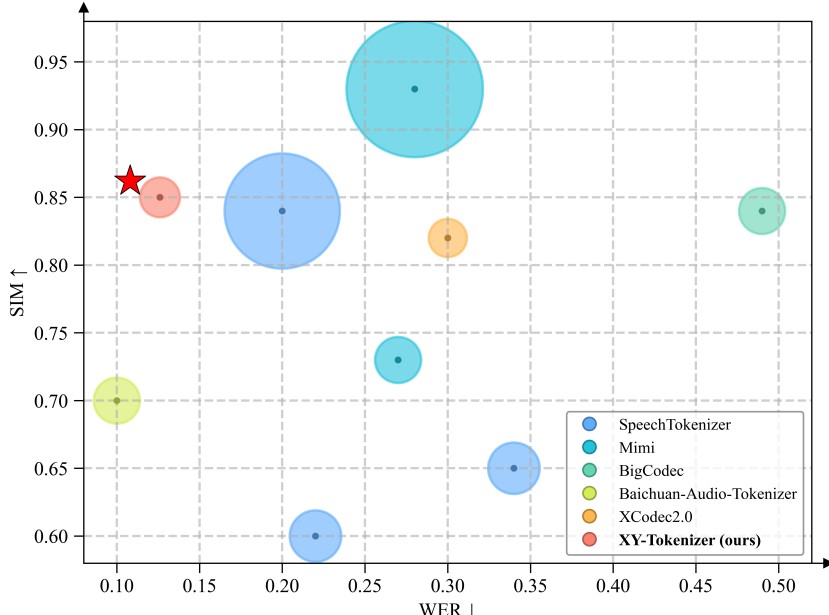

Figure 1: Comparison of speech codecs in semantic and acoustic performance. The horizontal axis shows word error rate (WER) of the automatic speech recognition probing task (Section 4.2), with lower values indicating better text alignment. The vertical axis measures audio reconstruction quality (Section 4.2). Circle size represents bitrate. Ideally, a speech codec should appear towards the top-left corner with a lower bitrate. Codec details are described in Section 4.1. The figure shows that **XY-Tokenizer** achieves strong semantic and acoustic performance at approximately 1 kbps, comparable to state-of-the-art codecs that typically excel in only one of the two aspects at a similar bitrate.

Moshi adopts a split residual vector quantization architecture, distilling one channel's output with a pretrained SSL model (Défossez et al., 2024). XCodec introduces an "X-shaped" structure, ensuring that tokens at each layer are semantically rich (Ye et al., 2025a). However, a key challenge in modeling both semantic and acoustic information **lies in the inherent conflict between these tasks, particularly at low bitrates, where achieving high performance in both remains difficult**.

In this work, we propose **XY-Tokenizer**, the first speech codec to successfully model both semantic and acoustic information effectively at low bitrates. Our codec employs a dual-tower architecture that **mitigates the conflict between semantic and acoustic tasks by minimizing shared parameters in a multi-task learning framework**. We introduce a multi-stage, multi-task training paradigm: the first stage aligns the codec with text using an LLM-based ASR approach and employs a reconstruction loss on the original speech signal to ensure coarse-grained audio reconstruction, utilizing a 2-channel encoder-decoder structure, forming an **X-shaped** architecture. The second stage incorporates a discriminator to model fine-grained audio features using a generative adversarial network (GAN) (Goodfellow et al., 2014). In this stage, the encoder and quantizer are kept fixed to maintain the alignment between speech tokens and text, while the decoder discards the text-alignment module, leading to a **Y-shaped** architecture.

Our contributions can be summarized as follows:

- We propose **XY-Tokenizer**, a speech codec with a 16kHz sampling rate and a 1kbps bitrate. It employs a dual-tower architecture to model semantic and acoustic information simultaneously through multi-task learning, aligns with text using an LLM-based automatic speech recognition approach, and ensures high-quality speech reconstruction via a codec decoder.

- We introduce a multi-stage, multi-task training paradigm for modeling semantic and acoustic information concurrently. The first stage aligns the codec with text and models coarse-grained audio features, while the second stage incorporates a discriminator to model fine-grained audio features using a generative adversarial network.

- We analyze the limitations of current speech codecs, particularly the inherent conflict between semantic and acoustic objectives, and propose solutions such as leveraging pretrained automatic speech recognition models and minimizing shared parameters to mitigate these conflicts.

- **XY-Tokenizer** achieves performance at 1kbps comparable to the 4kbps codec that simultaneously models semantic and acoustic information (e.g., **SpeechTokenizer**) in both semantic and acoustic dimensions. At similar low bitrates, it excels in both tasks and matches the performance of state-of-the-art codecs specialized in a single aspect, such as **BigCodec**, which models only acoustic quality without explicitly modeling semantic information. We conducted extensive ablation studies to validate the effectiveness of our approach, and we will open-source our repository and pretrained models.

## 2 PRELIMINARY EXPERIMENTS

Table 1: Comparison between pretrained ASR/SSL models in reconstructed audio quality; bold indicates best performance.

| Model | SIM ↑ | STOI ↑ | PESQ-NB ↑ | PESQ-WB ↑ |
|---|---|---|---|---|
| HuBERT | 0.42 | 0.80 | 1.46 | 1.20 |
| WavLM | 0.53 | 0.83 | 1.53 | 1.26 |
| Whisper | **0.68** | **0.88** | **2.03** | **1.65** |

Self-supervised learning (SSL) models for speech, such as those trained with masked language modeling (Hsu et al., 2021; Chen et al., 2022), effectively capture high-level speech features and are widely used in speech large language models (Speech LLMs) (Zhang et al., 2023a; 2024b). Similarly, automatic speech recognition (ASR) models, trained on large-scale paired speech–text datasets, achieve **strong alignment between speech and text modalities** (Ao et al., 2021; Tang et al., 2022; Radford et al., 2023). However, training a speech codec from scratch to align with the text modality is data-intensive. To address this, our proposed **XY-Tokenizer** leverages pretrained ASR or SSL models for the encoder to reduce training complexity. Although these ASR and SSL models exhibit strong alignment with text, **their ability to retain paralinguistic information remains underexplored**. To identify the most suitable pretrained model for our codec, we conduct a preliminary experiment to evaluate their performance in preserving acoustic information.

For this experiment, we selected three pre-trained models: Whisper (Radford et al., 2023), an ASR model, as well as HuBERT (Hsu et al., 2021) and WavLM (Chen et al., 2022), which are self-supervised learning (SSL) models. We trained an auto-encoder, which differs from a codec by removing the quantizer, to assess the reconstruction capabilities of these pre-trained models. Specifically, we used the pre-trained Whisper, HuBERT, and WavLM models as **fixed encoders**, each paired with a decoder of identical parameter size to ensure a fair comparison. The experimental setup and details are provided in Appendix A. As shown in Table 1, **Whisper achieves superior reconstruction performance**, effectively preserving paralinguistic information, such as speaker timbre and acoustic details. In contrast, HuBERT and WavLM exhibit limitations in preserving certain aspects of speaker timbre and fine-grained acoustic details. Furthermore, Whisper's pretraining on ASR tasks aligns closely with the LLM-based tasks employed in our codec, facilitating better speech-text alignment. Based on these findings, we selected Whisper to initialize the encoder of our proposed XY-Tokenizer and further fine-tuned it for our codec training pipeline.

## 3 METHOD

### 3.1 XY-TOKENIZER

**Motivation** An ideal speech codec should effectively balance two goals: high-fidelity audio reconstruction and strong semantic alignment with text (Zhang et al., 2023b; Yang et al., 2024b). However, these two objectives often conflict, as optimizing for one can degrade the other (Défossez et al., 2024). Our empirical analysis, as shown in Table 2, shows that decreasing the number of

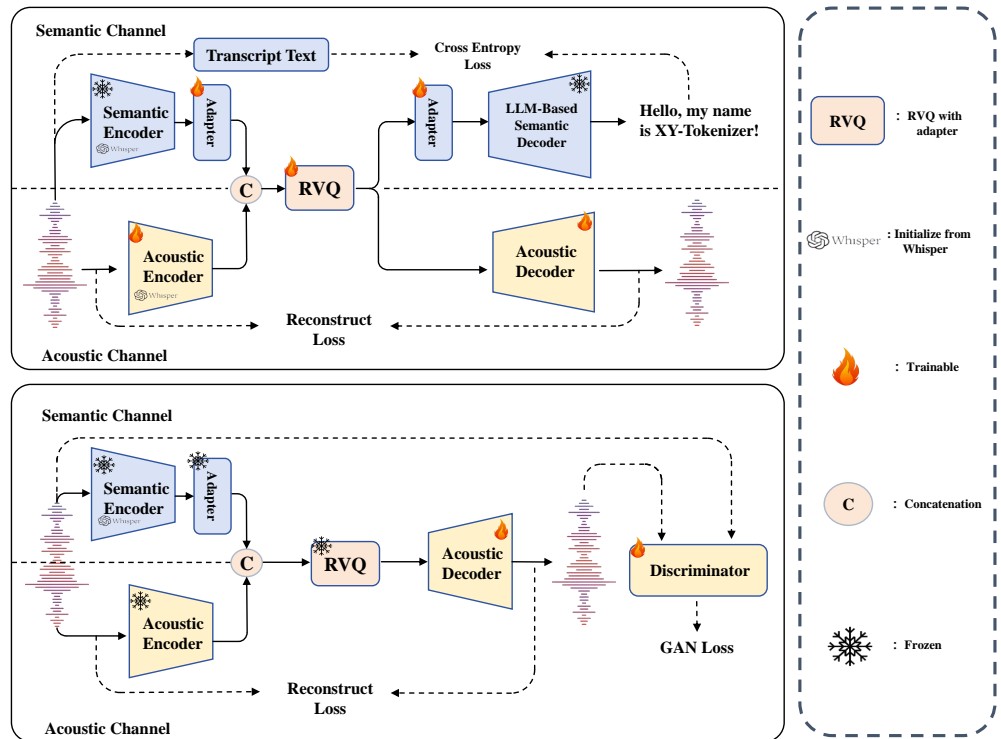

Figure 2: Illustration of **XY-Tokenizer**. The **upper half** depicts the **pre-training stage**, aligning **XY-Tokenizer** with text while preserving coarse acoustic features. The **lower half** illustrates the **post-training stage**, modeling finer-grained acoustic features. Model architecture and training procedure are detailed in Section 3.

Table 2: Impact of shared parameters on semantic and acoustic modeling performance. Details of the model architecture are provided in Appendix C. For Mimi-8, the shared parameters between the two tasks are the encoder and the semantic channel of the quantizer. WER is reported in the range $[0, 1]$.

| Model | Shared Parameters | SIM ↑ | WER ↓ |
|---|---|---|---|
| SpeechTokenizer-x1 | Encoder + Quantizer | 0.65 | 0.34 |
| Mimi-8 | Encoder | 0.73 | **0.28** |
| XCodec2.0 | Quantizer | **0.82** | 0.30 |

shared parameters between semantic and acoustic modeling pathways effectively mitigates the trade-off between high-fidelity audio reconstruction and strong semantic alignment. Moreover, semantic modeling can be effectively approached through automatic speech recognition (ASR) tasks, while acoustic modeling aligns closely with reconstruction through a codec decoder. To this end, we propose a dual-channel codec architecture that jointly models semantic and acoustic information in a multi-task setup, combining ASR and audio reconstruction, with shared parameters limited to the residual vector quantization (RVQ) module and its adjacent components.

**Encoder** The encoder comprises two parallel branches: a **semantic channel** and an **acoustic channel**, both processing mel-spectrogram inputs at 100 Hz. Each channel is initialized with a Whisper encoder (Radford et al., 2023), with the semantic encoder's parameters fixed and the acoustic encoder's parameters trainable. The semantic channel extracts linguistic features, while the acoustic channel captures paralinguistic information. The outputs of both channels are concatenated and further processed to produce the final encoder output. Additional details of the encoder architecture are provided in Appendix B.

**Quantizer** We employ a residual vector quantization (RVQ) module with 8 layers operating on the encoder output at a temporal resolution of 12.5 Hz (Zeghidour et al., 2021). Each layer uses a codebook of size 1024, resulting in a total bitrate of 1 kbps. The quantizer is integrated with adapter and convolution modules, with details provided in Appendix B.

**Decoder** The decoder consists of two parallel branches: a **semantic channel** and an **acoustic channel**, both processing the quantized encoder output. The semantic channel, a decoder-only large language model, generates text transcriptions, while the acoustic channel reconstructs the waveform. For details of the decoder, see Appendix B.

### 3.2 TWO STAGE TRAINING STRATEGY

To streamline the training process and enhance efficiency, we propose a two-stage training strategy, consisting of a pre-training stage and a post-training stage. In the pre-training stage, we employ multi-task learning to simultaneously model semantic features and coarse acoustic features. In the post-training stage, we focus on modeling fine-grained acoustic features. This section elaborates on these stages.

#### 3.2.1 PRE-TRAINING STAGE

In the pre-training stage, we focus on two tasks: audio reconstruction and automatic speech recognition (ASR). All model parameters are trainable, except for the weights of the semantic encoder, initialized from Whisper encoder (Radford et al., 2023), and the large language model (LLM) which is initialized from Qwen2.5 (Yang et al., 2024a). To align with text generation, we use the cross-entropy loss for the LLM, defined as:

$$\mathcal{L}_{asr} = -\sum_{t=1}^{N} \log p(\mathbf{y}_t \mid \mathbf{y}_{<t}, \mathbf{f}; \theta_{LLM})$$

where $\mathbf{y}_t$ is the predicted text token at time step $t$, $\mathbf{y}_{<t}$ denotes the sequence of preceding tokens, $\mathbf{f}$ represents the audio features input to the LLM, $N$ is the total number of predicted text tokens, and $\theta_{LLM}$ denotes the parameters of the LLM.

For modeling acoustic features, we employ a multi-scale mel-spectrogram reconstruction loss:

$$\mathcal{L}_{recon} = \sum_{i \in e} \|S_i(\mathbf{x}) - S_i(\hat{\mathbf{x}})\|_1$$

where $S_i$ is the mel-spectrogram at scale $i$, computed using a normalized short-time fourier transform (STFT) with a window size of $2^i$ and a hop length of $2^{i-2}$. The set of scales is defined as $e = \{5, \ldots, 11\}$. Here, $\mathbf{x}$ is the ground-truth audio waveform, and $\hat{\mathbf{x}}$ is the predicted waveform from the acoustic decoder. No waveform-based reconstruction loss is used.

Additionally, we incorporate a commitment loss to ensure effective quantization:

$$\mathcal{L}_{commit} = \sum_{i=1}^{N_q} \|\mathbf{z}_i - sg(q_i(\mathbf{z}_i))\|_1$$

where $\mathbf{z}_i$ is the input to the $i$-th layer of the quantizer, $q_i(\mathbf{z}_i)$ is its quantized output, $N_q$ is the number of quantized vectors, and $sg$ denotes the stop-gradient operation, which prevents gradients from propagating to the quantizer's codebook.

The total loss for the pre-training stage is a weighted combination of individual losses:

$$\mathcal{L}_{pretraining} = \lambda_{asr}\mathcal{L}_{asr} + \lambda_{recon}\mathcal{L}_{recon} + \lambda_{commit}\mathcal{L}_{commit}$$

where $\lambda_{asr}, \lambda_{recon}, \lambda_{commit}$ are hyperparameters that balance the weights of each loss term.

#### 3.2.2 POST-TRAINING STAGE

After the pre-training stage, we obtain an encoder capable of producing rich semantic features. However, the codec's output may contain artifacts, which significantly degrade perceptual quality

and listening experience. To address this, the post-training stage focuses on modeling fine-grained audio details. The approach is detailed below.

We adopt a generative adversarial network (GAN) framework for post-training. For the generator, which corresponds to the codec, we fix the encoder and quantizer to maintain the alignment between speech tokens and text, discard the semantic decoder used in the pre-training stage, and keep all other parameters consistent with the pre-training stage, with these parameters remaining trainable. For the discriminator, we employ multi-period discriminator (MPD), multi-scale discriminator (MSD), and multi-scale short-time fourier transform discriminator (MS-STFTD) to model higher-level features and improve the perceptual quality of the generated audio (Kong et al., 2020; Kumar et al., 2019; Défossez et al., 2022).

The discriminator loss follows the least squares GAN (LSGAN) formulation (Mao et al., 2017), given by:

$$\mathcal{L}_D(\mathbf{x}, \hat{\mathbf{x}}) = \frac{1}{K} \sum_{k=1}^{K} (1 - D_k(\mathbf{x}))^2 + D_k^2(\hat{\mathbf{x}})$$

where $D_k$ represents the $k$-th discriminator (from MPD, MSD, or MS-STFTD), $K$ is the total number of discriminators, $\mathbf{x}$ is the ground-truth audio, and $\hat{\mathbf{x}}$ is the predicted audio.

For the generator loss, we use the same multi-scale mel-spectrogram reconstruction loss as in the pre-training stage, denoted $\mathcal{L}_{recon}$. Additionally, we include a feature matching loss:

$$\mathcal{L}_{feat}(\mathbf{x}, \hat{\mathbf{x}}) = \frac{1}{KL} \sum_{k=1}^{K} \sum_{l=1}^{L} \frac{\left\| D_k^l(\mathbf{x}) - D_k^l(\hat{\mathbf{x}}) \right\|_1}{mean(\left\| D_k^l(\mathbf{x}) \right\|_1)} \tag{1}$$

where $D_k^l$ denotes the feature representation from the $l$-th layer of the $k$-th discriminator, $L$ is the number of layers per discriminator, and the mean is computed over all dimensions of $D_k^l(\mathbf{x})$. We also incorporate an adversarial loss:

$$\mathcal{L}_{adv}(\hat{\mathbf{x}}) = \frac{1}{K} \sum_{k=1}^{K} (1 - D_k(\hat{\mathbf{x}}))^2$$

The total generator loss is a weighted combination of these terms:

$$\mathcal{L}_G(\mathbf{x}, \hat{\mathbf{x}}) = \lambda_{recon}\mathcal{L}_{recon} + \lambda_{feat}\mathcal{L}_{feat} + \lambda_{adv}\mathcal{L}_{adv}$$

where $\lambda_{recon}, \lambda_{feat}, \lambda_{adv}$ are hyperparameters that balance the contributions of each loss term.

## 4 EXPERIMENTS

### 4.1 SETTINGS

**Dataset and Training Details**  We trained XY-Tokenizer using the full Emilia dataset, comprising approximately 101k hours of audio data, equivalent to about 37 million (audio, transcription) pairs (He et al., 2024). All audio data was resampled to 16 kHz. In the **pre-training stage**, audio clips longer than 30 seconds were truncated to the first 30 seconds, while clips shorter than 30 seconds were padded to 30 seconds, with loss computed only on the non-padded portions. We utilized 32 NVIDIA H100 GPUs, each with a batch size of 4, a maximum learning rate of $1 \times 10^{-4}$, and trained for 800,000 steps using DeepSpeed Zero2 (Rajbhandari et al., 2020). We used the AdamW optimizer with a weight decay of 0.01 (Loshchilov & Hutter, 2017). In the **post-training stage**, we randomly sampled 5-second segments from each audio clip for training, using a single NVIDIA H100 GPU with a batch size of 16. The generator was trained with a maximum learning rate of $1 \times 10^{-5}$, and the discriminator with a maximum learning rate of $1 \times 10^{-4}$, for 600,000 steps. For both the **pre-training stage** and **post-training stage**, we set $\lambda_{recon} = 15$. In the pre-training stage, we set $\lambda_{asr} = 20$ and $\lambda_{commit} = 1$. In the post-training stage, we set $\lambda_{feat} = 1$ and $\lambda_{adv} = 1$.

**Model Details**  We have detailed the codec architecture in Section 3.1 and Appendix B.

**Baselines**  We use SpeechTokenizer (Zhang et al., 2023b), Mimi (Défossez et al., 2024), XCodec2.0 (Ye et al., 2025b), and Baichuan Audio Tokenizer (Li et al., 2025) as our baseline

Table 3: Comparisons between different codecs in terms of semantic and acoustic performance on Librispeech dataset. WER refers to the word error rate measured on the ASR probing task, detailed in Section 4.2. Lower WER indicates better alignment with the original text content. WER is reported in the range $[0, 1]$; for example, a WER of 0.13 for XY-Tokenizer corresponds to a 13% word error rate on the ASR probing task. For codecs with >2k bps, bold values indicate SOTA performance; for low bitrate (∼1k bps) codecs, bold indicates the top 2 performing models. Baichuan refers to Baichuan Audio Tokenizer.

| | | | Semantic | | Acoustic | | | |
| Model | BPS | Frame Rate | Model Semantic | WER ↓ | SIM ↑ | STOI ↑ | PESQ-NB ↑ | PESQ-WB ↑ |
|---|---|---|---|---|---|---|---|---|
| DAC-8 | 6k | 75 | No | 0.74 | 0.88 | 0.95 | **3.79** | **3.46** |
| SpeechTokenizer | 4k | 50 | Yes | **0.20** | 0.84 | 0.92 | 3.05 | 2.60 |
| Mimi-32 | 4.4k | 12.5 | Yes | 0.28 | **0.93** | **0.96** | **3.79** | 3.42 |
| DAC-2 | 1.5k | 75 | No | 0.98 | 0.49 | 0.83 | 1.91 | 1.51 |
| BigCodec | 1.04k | 80 | No | 0.49 | **0.84** | **0.93** | **3.26** | **2.68** |
| SpeechTokenizer-x1 | 1.5k | 50 | Yes | 0.34 | 0.65 | 0.88 | 2.58 | 2.10 |
| SpeechTokenizer-x2 | 1.5k | 50 | Yes | 0.22 | 0.60 | 0.86 | 2.35 | 1.87 |
| SpeechTokenizer-x3 | 1.5k | 50 | Yes | 0.18 | 0.48 | 0.83 | 1.95 | 1.53 |
| Mimi-8 | 1.1k | 12.5 | Yes | 0.28 | 0.73 | 0.90 | 2.79 | 2.24 |
| Baichuan | 1.075k | 12.5 | Yes | **0.10** | 0.70 | 0.88 | 2.45 | 1.93 |
| XCodec2.0 | 0.8k | 50 | Yes | 0.30 | 0.82 | 0.91 | 3.03 | 2.43 |
| XY-Tokenizer(ours) | 1k | 12.5 | Yes | **0.13** | **0.85** | **0.92** | **3.10** | **2.50** |

codecs, which simultaneously model semantic and acoustic information. Details of these models are provided in Appendix C. Additionally, we include BigCodec (Xin et al., 2024), Descript Audio Codec (Kumar et al., 2023), which exclusively model acoustic information.

## 4.2 METRICS

**Reconstruction Evaluation** To evaluate the preservation of acoustic information, we employ several metrics. Speaker similarity (SIM) is calculated as the cosine similarity between speaker embeddings extracted from original and reconstructed audio using a pre-trained speaker verification model[1]. We also use short-time objective intelligibility (STOI) (Taal et al., 2010) to measure speech intelligibility and perceptual evaluation of speech quality (PESQ) (Rix et al., 2001) to assess audio quality. All evaluations were conducted on the LibriSpeech test-clean subset (Panayotov et al., 2015).

**Semantic Evaluation** To evaluate the semantic alignment between the codec and text, we employ an **automatic speech recognition (ASR) probing task**, adapted from the SUPERB framework (Yang et al., 2021), to assess the semantic quality of tokenized representations. We trained a downstream ASR model using **quantized embeddings**, with the pretrained codec fixed. These quantized embeddings are upsampled to a minimum frame rate of 50 Hz via replication before being fed into a downstream model (see Appendix D for details). The downstream model comprises a two-layer bidirectional LSTM optimized with CTC loss for character-level prediction (Hochreiter & Schmidhuber, 1997; Graves et al., 2006). All models are trained on the LibriSpeech train-clean-100 subset and evaluated on the LibriSpeech dev-clean subset (Panayotov et al., 2015), using word error rate (WER) as the metric for semantic performance, where a lower WER indicates better semantic alignment. The ASR probing task experiments are conducted with a batch size of 4, a maximum learning rate of $1 \times 10^{-4}$, and training for 400,000 steps.

## 4.3 EVALUATION RESULTS

As shown in Table 3, XY-Tokenizer achieves SOTA-comparable performance at similar bitrates, simultaneously excelling in both speech reconstruction and semantic preservation tasks.

**Speech Reconstruction** XY-Tokenizer achieves higher SIM scores than Mimi-8, SpeechTokenizer-RVQ-3 (including SpeechTokenizer-x1, SpeechTokenizer-x2, SpeechTokenizer-x3), and Baichuan

---

[1]https://github.com/microsoft/UniSpeech/tree/main/downstreams/speaker_verification

Table 4: Impact of reducing shared parameters between semantic and acoustic tasks. WER refers to word error rate on the automatic speech recognition probing task (detailed in Section 4.2). WER is reported in the range $[0, 1]$. Two Channel indicates whether the encoder simultaneously uses both semantic and acoustic channels.

| | | | | Semantic | Acoustic | | | |
|---|---|---|---|---|---|---|---|---|
| Model | Encoder | Two Channel | Encoder Params (M) | WER ↓ | SIM ↑ | STOI ↑ | PESQ-NB ↑ | PESQ-WB ↑ |
| (1) **XY-Tokenizer** | Whisper small | YES | 259 | 0.13 | **0.80** | **0.92** | **3.12** | **2.61** |
| (2) Single-channel | Whisper small | NO | 115 | 0.13 | 0.77 | **0.92** | 2.92 | 2.45 |
| (3) Single-channel | Whisper medium | NO | 356 | **0.12** | 0.77 | **0.92** | 2.88 | 2.38 |

Audio Tokenizer, with performance comparable to BigCodec, XCodec2.0, and SpeechTokenizer. These results provide preliminary evidence of the model's effectiveness in reconstructing speech. However, XY-Tokenizer's reconstruction quality is slightly inferior to high-bitrate codecs like DAC-8, likely due to greater information loss in low-bitrate codecs. We believe that low-bitrate codecs have significant potential for further enhancement in reconstruction performance.

**ASR Probing Results** XY-Tokenizer exhibits text alignment performance comparable to Baichuan Audio Tokenizer. XY-Tokenizer's word error rate (WER) on the ASR probing task is substantially lower than that of codecs employing representation distillation, such as SpeechTokenizer, Mimi-Codec, and XCodec 2.0. This performance may be attributed to language-based ASR tasks facilitating stronger text alignment compared to semantic distillation approaches, while also creating less conflict with reconstruction objectives. Conversely, codecs like DAC and BigCodec, which lack supervised text alignment during pretraining, exhibit higher WER in ASR probing tasks, likely due to a significant disparity between their compressed representations and textual content.

**Semantic-Acoustic Comprehensive Analysis** Considering both speech reconstruction and text alignment capabilities, our proposed XY-Tokenizer achieves excellent results in both aspects. While SpeechTokenizer performs well in both reconstruction and semantic tasks at higher bitrates, we observe that at lower bitrates, stronger distillation supervision leads to poorer reconstruction quality. This indicates that representation distillation methods can cause significant conflicts between semantic and acoustic learning objectives during codec training. Mimi-8 and XCodec2.0 demonstrate better reconstruction metrics at comparable bitrates than SpeechTokenizer, but perform less effectively on semantic tasks, which we attribute to the inherent trade-off between representation distillation and audio reconstruction objectives. Our proposed XY-Tokenizer achieves favorable results in both semantic and acoustic dimensions, suggesting that it effectively mitigates the conflict between these competing tasks to a noticeable extent. For ablation studies on conflict resolution approaches, please refer to Section 4.4.

**Additional Evaluations** We conducted further experiments to investigate the generalization capability of XY-Tokenizer. On cross-lingual and out-of-distribution datasets, XY-Tokenizer demonstrates superior or comparable performance in both text alignment and reconstruction compared to codecs operating at similar bitrates. Moreover, in LLM-based understanding and generation tasks, XY-Tokenizer consistently outperforms codecs with comparable bitrates, such as Mimi-8. Detailed results are provided in Appendix E and Appendix F.

## 4.4 ABLATION STUDY

To evaluate the effectiveness of our proposed XY-Tokenizer in simultaneously modeling semantic and acoustic features while mitigating conflicts between these tasks, we conducted a series of ablation experiments. Unless otherwise specified, all ablation experiments were performed during the pre-training phase without a post-training stage, utilizing the same dataset and preprocessing methods as described in Section 4.1. We employed a global batch size of 128, a maximum learning rate of $1 \times 10^{-4}$, and trained for 200,000 steps, with all loss weights consistent with those outlined in Section 4.1.

**Shared Parameters Cause Conflicts** In XY-Tokenizer, parameters between semantic and acoustic tasks are shared only in the residual vector quantization (RVQ) module and its adjacent components. To assess the effectiveness of minimizing shared parameters in reducing semantic-acoustic conflicts, we trained three models: (1) the proposed XY-Tokenizer, (2) XY-Tokenizer without a semantic encoder, where the encoder is a single-channel, trainable acoustic encoder, and (3) XY-Tokenizer

Table 5: Impact of LLM trainability on model performance over training steps. LLM WER refers to the WER of text decoded by the LLM-based semantic decoder during pretraining. Probing WER refers to the word error rate measured on the ASR probing task(detailed in Section 4.2). Both LLM WER and Probing WER are reported in the range $[0, 1]$.

| Model | Train Steps | Semantic | | Acoustic | | | |
| | | Probing WER ↓ | LLM WER ↓ | SIM ↑ | STOI ↑ | PESQ-NB ↑ | PESQ-WB ↑ |
| --- | --- | --- | --- | --- | --- | --- | --- |
| (1) Fixed LLM | 200K | 0.13 | 0.06 | 0.80 | 0.92 | 3.12 | 2.61 |
| | 400K | 0.13 | 0.05 | 0.81 | 0.93 | 3.22 | 2.67 |
| | 600K | 0.14 | 0.05 | 0.83 | 0.93 | 3.30 | 2.78 |
| | 800K | **0.13** | 0.05 | **0.84** | 0.93 | 3.35 | 2.83 |
| (2) Trainable LLM | 200K | 0.18 | **0.03** | 0.80 | 0.93 | 3.21 | 2.69 |
| | 400K | 0.20 | 0.04 | 0.82 | 0.93 | 3.32 | 2.77 |
| | 600K | 0.22 | **0.03** | 0.83 | **0.94** | 3.41 | 2.88 |
| | 800K | 0.24 | **0.03** | **0.84** | **0.94** | **3.46** | **2.96** |

without a semantic encoder, with the acoustic encoder initialized using Whisper-medium weights, maintaining a parameter count comparable to (1). In models (2) and (3), the semantic and acoustic tasks share both the encoder and RVQ modules. As shown in Table 4, all three models perform well on the ASR probing task, demonstrating that leveraging an LLM-based ASR approach enables the codec to effectively align with textual semantics. However, models (2) and (3) exhibit inferior performance on reconstruction metrics compared to (1), despite model (3) having a similar parameter count to (1). This suggests that **sharing model parameters across semantic and acoustic tasks is a primary cause of task conflict**. Thus, reducing shared parameters is an effective strategy for simultaneously achieving high-quality audio reconstruction and improved text alignment.

**Fixing LLM for Enhanced Training Stability** In the pre-training stage of our XY-Tokenizer, the pretrained LLM is kept fixed. To explore whether allowing the LLM to be trainable could yield better results, we trained two models: (1) the proposed XY-Tokenizer with a fixed LLM, and (2) XY-Tokenizer with a trainable LLM. Table 5 indicates that both models achieve comparable audio reconstruction quality. On the Word Error Rate (WER) of text decoded by the LLM, model (2) outperforms model (1) with a lower WER. However, on the ASR probing task, model (2) performs worse than model (1). Moreover, as training progresses, the performance of model (2) on the ASR probing task deteriorates, while model (1) remains stable. We hypothesize that **allowing the LLM to be trainable increases the flexibility of the semantic decoder, causing the encoder's text-alignment capability to gradually shift toward the semantic decoder**. This shift likely explains the discrepancy between the ASR probing task and LLM decoder WER performance in models (1) and (2). We conclude that fixing the LLM is a more appropriate approach, offering stability and effectively balancing semantic and acoustic modeling.

**Additional Ablation Studies** We conduct further ablations on training strategies and model architectures, with detailed results provided in Appendix G.

## 5 RELATED WORK

Related work about speech large language models and neural speech codecs is provided in Appendix H.

## 6 CONCLUSION

In this study, we propose XY-Tokenizer, a novel codec that mitigates the conflict between understanding and generation capabilities through multi-task learning, enhances reconstruction quality, and improves text alignment at low bitrates. Experimental results demonstrate that XY-Tokenizer excels in both speech reconstruction and understanding tasks. We conducted ablation studies to optimize the modeling of semantic and acoustic information. Please refer to the appendix for additional experimental details.

ETHICS STATEMENT

This work complies with the ICLR Code of Ethics. No human or animal subjects were involved, and no personally identifiable information was used. The research poses no privacy or security concerns, and we are committed to transparency and integrity throughout the process.

REPRODUCIBILITY STATEMENT

We have provided detailed descriptions of the experimental setup, including training procedures, model configurations, and evaluation methods, to facilitate reproducibility of our results.

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

## A   PRELIMINARY EXPERIMENT SETTINGS

**Model Architecture**   For the preliminary experiments to select the encoder for the XY-Tokenizer, we utilized three pre-trained models: whisper-small[2], hubert-large-ll6k[3], and wavlm-large[4]. **These encoders were kept frozen during training to evaluate their reconstruction capabilities**. A decoder with approximately 250M parameters was used, without a quantizer.

**Dataset and Training**   The auto-encoders were trained on the full Emilia dataset. Training was performed on a single NVIDIA H100 GPU with a batch size of 8. The training process consisted of 200,000 steps. The maximum learning rate was set to $1 \times 10^{-4}$, and DeepSpeed Zero-2 was used for optimization.

---

[2]https://huggingface.co/openai/whisper-small
[3]https://huggingface.co/facebook/hubert-large-ll60k
[4]https://huggingface.co/microsoft/wavlm-large

# B  MODEL DETAILS

## B.1  ADAPTER

To enhance the flexibility of embeddings, we incorporate lightweight Transformer-based (Vaswani et al., 2017) adapter modules at multiple components of the XY-Tokenizer. Each adapter consists of a 4-layer Transformer with a hidden dimension of 768, a feed-forward network (FFN) dimension of 3072, and 12 attention heads. Adapters are placed after the semantic encoder, before and after the quantizer, and before the LLM-based semantic decoder.

## B.2  ENCODER

The input waveform is resampled to 16 kHz, and an 80-channel Mel spectrogram is computed using a 25 ms window length and a 10 ms hop length to serve as the input to the encoder.

The semantic encoder adopts the whisper-small encoder configuration and processes the Mel spectrogram through the following modules: (1) a 1D convolutional layer with a kernel size of 3 and a stride of 1, projecting the 80-dimensional input to a hidden dimension of 768; (2) a GELU activation function (Hendrycks & Gimpel, 2016); (3) a second 1D convolutional layer with a kernel size of 3 and a stride of 2, reducing the sequence length by a factor of 2; (4) another GELU activation function; (5) sinusoidal positional embeddings; (6) a transformer with 12 layers, 12 attention heads, a dimension of 768, and a feed-forward network dimension of 3072. The semantic encoder's output is then passed to (7) an adapter module (detailed in Appendix B.1). The semantic encoder's parameters are fixed during training.

The acoustic encoder follows a similar architecture to the semantic encoder but is trainable and excludes the adapter module. The outputs of the semantic and acoustic encoders are concatenated along the feature dimension.

## B.3  QUANTIZER

We employ a residual vector quantizer (RVQ) with 8 layers and a codebook size of 1024 per layer. The codebook is updated using an exponential moving average (EMA) with a weight decay of 0.99. To prevent codebook collapse, unused codebook entries are randomly replaced with input vectors from the current batch after several training steps. The codebook is initialized using $k$-means clustering with 10 iterations. A $4\times$ downsampling convolutional layer is applied before the quantizer, reducing the encoder's 50 Hz embeddings to 12.5 Hz, resulting in a bitrate of 1 kbps for our proposed XY-Tokenizer. Adapter modules (detailed in Appendix B.1) are placed before the downsampling convolution and after the quantizer.

## B.4  DECODER

The decoder processes quantized features through two distinct pathways: the semantic decoder for text prediction and the acoustic decoder for audio reconstruction.

The semantic decoder takes the output of the quantizer as input, passes it through an adapter (detailed in Appendix B.1), and uses the resulting features as conditioning input for a decoder-only large language model (LLM). The LLM, based on Qwen2.5-0.5B (Yang et al., 2024a), has a hidden dimension of 896, an intermediate layer size of 4864, and 24 layers, generating the final predicted text corresponding to the input speech.

The acoustic decoder takes the output of the quantizer as input, applies a 4x upsampling convolution to reach 50 Hz, and follows a structure symmetric to the acoustic encoder to achieve 100 Hz. Finally, a 30-layer Vocos model (Siuzdak, 2023) with a hop size of 160 reconstructs the 16 kHz audio waveform.

**Discriminators**  To ensure high perceptual quality, we employ three discriminator models: multi-period discriminator (MPD)  (Kong et al., 2020), multi-scale discriminator (MSD) (Kumar et al., 2019), and multi-scale short-time fourier transform discriminator (MS-STFTD) (Défossez et al., 2022). The parameters of our discriminator models are consistent with those used in SpeechTokenizer  (Zhang et al., 2023b).

## C  BASELINE MODEL DETAILS

In this section, we provide detailed descriptions of the baseline models used in our experiments. For **Mimi**, we adopt the official RVQ-8 and RVQ-32 versions, referred to as Mimi-8 and Mimi-32, respectively. For **XCodec 2.0**, **Baichuan Audio Tokenizer**, and **BigCodec**, we use the official checkpoints provided by their respective authors.

For **Descript Audio Codec (DAC)**, we utilize the official implementation with a sampling rate of 24 kHz and residual vector quantization (RVQ) levels of 2 and 8, denoted as DAC-2 and DAC-8.

For **SpeechTokenizer**, we employ the official `speechtokenizer_hubert_avg` version [5]. Additionally, we train three variants of SpeechTokenizer using the official codebase, modifying only the RVQ layers and the distillation weight (`distill_loss_lambda`). Specifically, we reduce the RVQ layers from 8 to 3 and train three versions: (1) RVQ-3 with `distill_loss_lambda` = 24 ($5\times$ smaller than the official setting), denoted as SpeechTokenizer-x1, (2) RVQ-3 with `distill_loss_lambda` = 120 (matching the official setting), denoted as SpeechTokenizer-x2, and (3) RVQ-3 with `distill_loss_lambda` = 600 ($5\times$ larger than the official setting), denoted as SpeechTokenizer-x3.

---

[5]`https://huggingface.co/fnlp/SpeechTokenizer/tree/main/speechtokenizer_hubert_avg`

# D    ASR PROBING TASK DETAILS

To enable effective alignment in the automatic speech recognition (ASR) probing task, particularly for low-bitrate codecs, **we upsample the embeddings for models with a frame rate below 50 Hz to a minimum of 50 Hz using replication**. This upsampling is necessary because an insufficient input sequence length ($T$) relative to the target sequence length ($U$) can prevent the connectionist temporal classification (CTC) loss from effectively aligning the input sequence (quantized features) with the target sequence (transcription characters). Specifically, CTC requires $T \geq U$ to accommodate at least one time step per target label, and in the worst case, $T \geq 2U + 1$ to account for potential blank labels between each target label and at the sequence boundaries. Upsampling ensures that $T$ is sufficiently large, particularly for low-frame-rate codes, to satisfy these constraints and enable effective alignment.

# E   EVALUATING XY-TOKENIZER'S GENERALIZATION CAPABILITIES

Table 6: Comparisons between different codecs in terms of semantic and acoustic performance in **cross-lingual scenarios**. WER refers to the word error rate measured on the ASR probing task, detailed in Section 4.2. Lower WER indicates better alignment with the original text content. WER is reported in the range $[0, 1]$. For codecs with $>2k$ bps, bold values indicate SOTA performance; for low bitrate ($\sim$1k bps) codecs, bold indicates the top 2 performing models. Baichuan refers to Baichuan-Audio-Tokenizer.

| Model | BPS | Frame Rate | Semantic | | Acoustic | | | |
| | | | Model Semantic | WER ↓ | SIM ↑ | STOI ↑ | PESQ-NB ↑ | PESQ-WB ↑ |
|---|---|---|---|---|---|---|---|---|
| DAC-8 | 6k | 75 | No | 0.95 | 0.93 | 0.94 | **3.74** | **3.41** |
| SpeechTokenizer | 4k | 50 | Yes | **0.68** | 0.89 | 0.90 | 2.96 | 2.53 |
| Mimi-32 | 4.4k | 12.5 | Yes | 0.78 | **0.95** | **0.95** | 3.68 | 3.26 |
| DAC-2 | 1.5k | 75 | No | 1.00 | 0.60 | 0.82 | 1.79 | 1.42 |
| BigCodec | 1.04k | 80 | No | 0.85 | **0.88** | **0.92** | **3.14** | **2.55** |
| SpeechTokenizer-x1 | 1.5k | 50 | Yes | 0.78 | 0.75 | 0.88 | 2.56 | 2.11 |
| SpeechTokenizer-x2 | 1.5k | 50 | Yes | 0.69 | 0.70 | 0.85 | 2.29 | 1.84 |
| SpeechTokenizer-x3 | 1.5k | 50 | Yes | 0.67 | 0.56 | 0.80 | 1.86 | 1.48 |
| Mimi-8 | 1.1k | 12.5 | Yes | 0.77 | 0.80 | 0.89 | 2.66 | 2.13 |
| Baichuan | 1.075k | 12.5 | Yes | **0.52** | 0.75 | 0.84 | 2.13 | 1.68 |
| XCodec2.0 | 0.8k | 50 | Yes | 0.75 | **0.87** | **0.90** | **2.93** | **2.36** |
| XY-Tokenizer(ours) | 1k | 12.5 | Yes | **0.47** | 0.86 | 0.89 | 2.61 | 2.09 |

Table 7: Comparisons between different codecs in terms of semantic and acoustic performance in **out-of-domain (OOD) scenarios**. WER refers to the word error rate measured on the ASR probing task, detailed in Section 4.2. Lower WER indicates better alignment with the original text content. WER is reported in the range $[0, 1]$. For codecs with $>2k$ bps, bold values indicate SOTA performance; for low bitrate ($\sim$1k bps) codecs, bold indicates the top 2 performing models. Baichuan refers to Baichuan-Audio-Tokenizer.

| Model | BPS | Frame Rate | Semantic | | Acoustic | | | |
| | | | Model Semantic | WER ↓ | SIM ↑ | STOI ↑ | PESQ-NB ↑ | PESQ-WB ↑ |
|---|---|---|---|---|---|---|---|---|
| DAC-8 | 6k | 75 | No | 0.80 | 0.91 | 0.94 | 3.69 | 3.22 |
| SpeechTokenizer | 4k | 50 | Yes | **0.38** | 0.83 | 0.90 | 2.82 | 2.31 |
| Mimi-32 | 4.4k | 12.5 | Yes | 0.41 | **0.94** | **0.95** | **3.77** | **3.26** |
| DAC-2 | 1.5k | 75 | No | 0.97 | 0.54 | 0.82 | 1.83 | 1.42 |
| BigCodec | 1.04k | 80 | No | 0.56 | 0.84 | **0.92** | **3.06** | **2.46** |
| SpeechTokenizer-x1 | 1.5k | 50 | Yes | 0.52 | 0.65 | 0.87 | 2.41 | 1.93 |
| SpeechTokenizer-x2 | 1.5k | 50 | Yes | 0.44 | 0.59 | 0.84 | 2.16 | 1.72 |
| SpeechTokenizer-x3 | 1.5k | 50 | Yes | 0.42 | 0.47 | 0.80 | 1.78 | 1.41 |
| Mimi-8 | 1.1k | 12.5 | Yes | 0.41 | 0.79 | 0.90 | 2.80 | 2.25 |
| Baichuan | 1.075k | 12.5 | Yes | **0.19** | 0.70 | 0.86 | 2.29 | 1.82 |
| XCodec2.0 | 0.8k | 50 | Yes | 0.43 | **0.85** | **0.91** | 2.94 | 2.39 |
| XY-Tokenizer(ours) | 1k | 12.5 | Yes | **0.25** | **0.87** | **0.91** | **2.99** | **2.49** |

We conducted the following experiments to investigate the generalization capabilities of XY-Tokenizer in more complex scenarios.

**Cross-Lingual Performance**   We randomly selected 100 hours of audio from the MLS-Dutch(Pratap et al., 2020) train-subset as our training set for the ASR probing task mentioned in Section 4.2. We then tested the WER and reconstruction metrics on the dev-clean dataset. As shown in Table 6, we found that XY-Tokenizer performed best on the ASR probing task, which we attribute to the generalization capabilities of our LLM-based ASR task across languages. Regarding reconstruction metrics, among low-BPS codecs, BigCodec, Mimi-8, XCodec2.0, and XY-Tokenizer all performed well. However, XY-Tokenizer was slightly inferior to XCodec2.0, which we hypothesize

is because XCodec2.0's training set included MLS, whereas XY-Tokenizer's did not. The reconstruction metrics of low-BPS codecs still lag behind high-BPS codecs, a gap we aim to bridge with better methods in the future.

**Out-of-Distribution Data** We randomly selected 100 hours of audio from the VoxPopuli-EN(Wang et al., 2021) train-subset as our training set for the ASR probing task mentioned in Section 4.2. We then tested the WER and reconstruction metrics on the dev-clean dataset. As shown in Table 7, we found that XY-Tokenizer achieved strong results on both the ASR probing task and reconstruction metrics, demonstrating its generalization ability on out-of-distribution datasets.

# F EVALUATING XY-TOKENIZER'S PERFORMANCE ON LLM-BASED UNDERSTANDING AND GENERATION TASKS

We conducted experiments on XY-Tokenizer and other speech codecs with similar experimental settings, using LLM-based understanding and generation tasks with discrete tokens as input.

## F.1 LLM-BASED UNDERSTANDING TASK

Table 8: Performance of different codecs on the Librispeech dataset in an LLM-based understanding task. WER is calculated on the text decoded from the LLM and the ground truth transcription. WER is reported in the range $[0, 1]$. Lower WER indicates better alignment with the original text content.

| **Model** | WER-dev-clean | WER-test-clean | WER-test-other |
|-----------|---------------|----------------|----------------|
| Mimi-8 | 0.096 | 0.096 | 0.242 |
| XCodec2.0 | 0.138 | 0.141 | 0.324 |
| XY-Tokenizer | **0.055** | **0.059** | **0.116** |

We trained a decoder-only LLM-based ASR model using Qwen3 0.6B (Yang et al., 2025) as the base and the Librispeech train subset (train-clean-100, train-clean-360, and train-other-500, a total of 960 hours) as the training data. We use **discrete tokens of codec as LLM's input and the LLM is trained to autoregressively predict text tokens**.

We evaluated the model on the Librispeech dev-clean, test-clean, and test-other subsets, using Word Error Rate (WER) to measure the alignment between the speech codec's tokens and text, as well as its capabilities in LLM-based understanding.

As shown in Table 8, XY-Tokenizer demonstrates a significantly lower WER than Mimi-8 and XCodec2.0. This indicates that **the speech tokens of XY-Tokenizer align better with text**, making it more suitable as a speech tokenizer for Speech LLMs.

## F.2 LLM-BASED GENERATION TASK

Table 9: Performance of different codecs on zero-shot TTS task.

| Model | test-clean | | | test-other | | |
|-------|---------|-------|--------|---------|-------|--------|
| | WER↓ | SIM↑ | UTMOS↑ | WER↓ | SIM↑ | UTMOS↑ |
| Mimi-8 | 0.109 | 0.44 | 3.38 | 0.1708 | 0.38 | 2.96 |
| XY-Tokenizer | **0.107** | **0.51** | **4.01** | **0.1272** | **0.47** | **3.55** |

We trained a Text-to-Speech (TTS) model using XY-Tokenizer and other speech codecs to evaluate their performance on a zero-shot TTS task. For training, we used Qwen3 0.6B (Yang et al., 2025) as the base and the Emilia dataset as the training set to train **a purely autoregressive (AR)** LLM-based Text-to-Speech (TTS) model with a delay interleaving pattern as mentioned in MusicGen (Copet et al., 2023).

For evaluation, we randomly selected 100 audio clips between 3 and 10 seconds from the Librispeech test-clean and test-other subsets to serve as prompts. For each prompt, we randomly selected another utterance by the same speaker. We then concatenated the prompt text, the text to be generated, and the prompt speech and fed this into the speech LLM to synthesize the audio. We used Speaker Similarity, WER (transcribed by Whisper large v3), and UTMOS (Saeki et al., 2022) to evaluate the performance of the TTS model.

As shown in Table 9, XY-Tokenizer outperforms Mimi-8 on the zero-shot TTS task, indicating that **XY-Tokenizer is well-suited for speech LLM modeling**. This further demonstrates the effectiveness of our proposed methods: mitigating semantic-acoustic conflicts by reducing shared parameters, implementing a multi-stage, multi-task training strategy, and leveraging a pretrained automatic speech recognition models.

# G ADDITIONAL ABLATIONS

## G.1 TRAINING FROM WHISPER FOR REDUCED TRAINING COMPLEXITY

Table 10: Effectiveness of pretrained Whisper weights in mitigating semantic-acoustic conflicts. WER refers to word error rate on the ASR probing task(lower is better, detailed in Section 4.2). WER is reported in the range $[0, 1]$.

| | Semantic | Acoustic | | | |
|---|---|---|---|---|---|
| Model | WER ↓ | SIM ↑ | STOI ↑ | PESQ-NB ↑ | PESQ-WB ↑ |
| (1) With Whisper weights | **0.13** | 0.77 | 0.92 | 2.92 | 2.45 |
| (2) Without Whisper weights | 0.27 | **0.82** | **0.93** | **3.15** | **2.60** |

To investigate the role of pre-trained whisper weights in mitigating semantic-acoustic conflicts, we trained two models: (1) a model with an acoustic encoder initialized with whisper-small weights, without a semantic encoder, and (2) the same model architecture as (1), but without loading whisper-small weights. As shown in Table 10, model (1) outperforms model (2) in the ASR probing task. This result highlights that whisper, pre-trained on extensive supervised data, is effective in alleviating semantic-acoustic conflicts in speech codec.

## G.2 IMPORTANCE OF LLM-BASED ASR TASK IN FINE-TUNING WHISPER

In the pre-training stage, we adopt a multi-task learning approach to jointly model semantic and acoustic information. Specifically, we leverage a language model-based automatic speech recognition (ASR) task to align the codec's quantized representations with text, while the codec's decoder preserves paralinguistic information. The encoder is initialized with pretrained Whisper weights and fine-tuned during training. Given that Whisper is pretrained on ASR tasks and demonstrates strong text alignment, we investigate whether the LLM-based ASR task further enhances the encoder's alignment with text. To this end, we conduct the following ablation study.

We train two models: (1) A modified version of our proposed XY-Tokenizer, where the encoder includes only the acoustic channel and omits the semantic channel, with all other components unchanged; (2) A variant of (1) that excludes the LLM-based ASR supervision, relying solely on reconstruction loss and commitment loss. As shown in Table 11, both models exhibit strong audio reconstruction capabilities. However, the word error rate (WER) on the ASR probing task reveals that model (2) performs significantly worse in text alignment compared to model (1). These results underscore the critical role of the LLM-based ASR task in enhancing text alignment, enabling XY-Tokenizer to optimize both text alignment and audio reconstruction effectively when combined with the reconstruction task.

## G.3 CHOICE OF ACOUSTIC DECODER

We conducted an ablation study on the acoustic decoder. We trained a 12.5Hz RVQ-8 autoencoder (consisting of only an acoustic encoder, quantizer, and acoustic decoder, with no semantic encoder or decoder, and only the pre-training stage) for 180k steps. We used three models as the acoustic decoder: Vocos (used in XY-Tokenizer), SEANet decoder (Tagliasacchi et al., 2020) (also used in Mimi Codec), and HiFi-GAN vocoder (Kong et al., 2020).

Based on the results detailed in Table 12, Vocos demonstrated superior reconstruction performance compared to both HiFi-GAN and SEANet. This finding validates the effectiveness of our method in generating high-quality speech.

## G.4 NECESSITY OF TWO-STAGE TRAINING STRATEGY

We adopt a two-stage training strategy. In the **pre-training stage**, the encoder and quantizer of the XY-Tokenizer are optimized through an ASR task to align their representations with text, while a reconstruction task is employed to capture coarse-grained acoustic features. In the **post-training**

Table 11: Effectiveness of the LLM-based ASR task for fine-tuning Whisper. WER refers to word error rate on the ASR probing task(lower is better, detailed in Section 4.2). WER is reported in the range $[0, 1]$.

| Model | Semantic | Acoustic | | | |
|---|---|---|---|---|---|
| | WER ↓ | SIM ↑ | STOI ↑ | PESQ-NB ↑ | PESQ-WB ↑ |
| (1) With ASR Supervision | **0.13** | 0.77 | 0.92 | 2.92 | 2.45 |
| (2) Without ASR Supervision | 0.58 | **0.82** | **0.93** | **3.26** | **2.74** |

Table 12: Comparison of various acoustic decoders.

| Model | SIM↑ | STOI↑ | PESQ-NB↑ | PESQ-WB↑ |
|---|---|---|---|---|
| Vocos | **0.82** | **0.93** | **3.23** | **2.69** |
| HiFi-GAN | 0.81 | 0.92 | 3.07 | 2.55 |
| SEANet | **0.82** | **0.93** | 3.11 | 2.58 |

Table 13: Comparison of training efficiency between the proposed **two-stage training strategy (pre-train + post-train)** and the single-stage approach (e.g., SpeechTokenizer, DAC). Encoder Trainable indicates whether the acoustic encoder is trainable. Decoder Trainable indicates whether the acoustic decoder is trainable. With LLM denotes whether a semantic decoder (based on Qwen2.5 LLM) is included. Throughput is measured as the amount of audio (in seconds) processed per GPU per second during training. The results show that **the throughput of the single-stage approach is significantly lower than that of the our proposed two-stage strategy**.

| Stage | Encoder Trainable | Decoder Trainable | With LLM | With Discriminator | Throughput |
|---|---|---|---|---|---|
| Single Stage | Yes | Yes | Yes | Yes | 40.8 |
| Pre-train Stage | Yes | Yes | Yes | No | 136.7 |
| Post-train Stage | No | Yes | No | Yes | 46.6 |

**stage**, we freeze the encoder and quantizer to **preserve the text-token alignment ability** of the XY-Tokenizer, and introduce a discriminator to model fine-grained acoustic information. This design choice is motivated by the following considerations:

(1) **Training stability** From our empirical experiments, we found that the RVQGAN structure becomes unstable at low bitrates (approximately $\leq 1.5$ kbps). Removing the discriminator during pre-training significantly improved stability.

(2) **Training efficiency** As shown in Section 4 and Table 13, during the pre-training stage we use a batch size of 4, with each audio padded to 30 seconds (same as Whisper), resulting in 120 seconds of audio per GPU. In the post-training stage, we use a batch size of 16, with each audio clipped to 5 seconds, resulting in 80 seconds of audio per GPU. In both stages, GPU memory utilization is already close to its maximum capacity. We further observe that **incorporating the discriminator in the post-training stage substantially increases computational cost and reduces throughput during training**. If the two stages were merged, the batch size would need to be reduced even further, resulting in much lower training efficiency.

Therefore, we employ the two-stage training strategy to balance stability, efficiency, and modeling capability.

# H RELATED WORK

## H.1 SPEECH LANGUAGE MODELS

Recent research on speech large language models has attracted considerable interest (Latif et al., 2023; Wu et al., 2024; Ji et al., 2024). AudioLM (Borsos et al., 2023) achieves high-quality audio generation with coherent long-term structure through coarse-to-fine token modeling. SpeechGPT (Zhang et al., 2023a), the first end-to-end speech large language model, features strong instruction-following capabilities and effective spoken dialogue interaction, employing a three-stage training methodology to facilitate cross-modal transfer and efficient training. SpeechGPT-Gen (Zhang et al., 2024a) proposes Chain-of-Information Generation, a modeling approach that disentangles semantic and perceptual aspects for large-scale speech generation. IntrinsicVoice (Zhang et al., 2024b) implements GroupFormer to diminish the modality gap between text and speech, thereby enabling the transfer of capabilities from pre-trained large language models to the speech domain, facilitating low-latency and high-quality speech interaction in multi-turn dialogue contexts. Moshi (Défossez et al., 2024) employs a multi-stream architecture that concurrently processes audio streams from both the user and the system (Moshi itself), supporting dynamic conversations with overlaps and interruptions, thereby achieving full-duplex dialogue.

## H.2 SPEECH CODECS

Speech codecs play a vital role in speech large language models by converting continuous speech signals into discrete tokens, enabling LLMs to process speech as a form of "foreign language." Neural network-based speech codecs predominantly utilize the RVQGAN paradigm, which can compress audio signals into low-bitrate representations through end-to-end training (Zeghidour et al., 2021; Défossez et al., 2022; Kumar et al., 2023), making them ideal for real-time communication applications. BigCodec (Xin et al., 2024) achieves excellent reconstruction quality even at low bitrates by scaling the encoder and decoder parameters. To align speech codec tokens with large text models, recent efforts have explored modeling both semantic and acoustic features simultaneously (Zhang et al., 2023b; Défossez et al., 2024; Ye et al., 2025a). SpeechTokenizer (Zhang et al., 2023b) enhances the RVQGAN paradigm with semantic distillation to guide the first layer of RVQ to align with a teacher SSL model (Hsu et al., 2021). X-Codec (Ye et al., 2025a) proposes an X-shaped structure where each layer of RVQ contains both semantic and acoustic information. Baichuan Audio Tokenizer (Li et al., 2025) first obtains a coarse Mel-spectrogram through multi-task learning and text alignment, then generates an enhanced Mel-spectrogram via conditional flow matching (Lipman et al., 2022), which is finally converted into waveforms using a pretrained vocoder (Kong et al., 2020).

# I  USE OF LLMS

In this work, we used a large language model (LLM) to assist with language polishing and improving the clarity of writing.

## J LIMITATIONS

In this paper, we propose XY-Tokenizer, a codec designed to model both semantic and acoustic information while effectively mitigating conflicts between these tasks. Experimental results demonstrate that XY-Tokenizer achieves strong performance in audio reconstruction quality and text alignment. However, several limitations remain. First, achieving lower bitrates without compromising performance remains a challenge. Additionally, the scaling law for training speech codecs, particularly with respect to parameter count and dataset size, requires further investigation to optimize training efficiency and generalization.

