# OpenReview forum: "XY-Tokenizer: Mitigating the Semantic-Acoustic Conflict in Low-Bitrate Speech Codecs"
_ICLR.cc/2026/Conference — ICLR 2026 Conference Withdrawn Submission_

### Official Review · Reviewer_6qgE · 2025-10-29

**Soundness:** 2
**Presentation:** 2
**Contribution:** 2
**Rating:** 2
**Confidence:** 4

**Summary:**

This paper presents XY-Tokenizer, which is a speech codec designed to jointly model semantic and acoustic information. To this end, XY-Tokenizer consists of a dual-channel encoder (frozen semantic channel with a trainable acoustic channel), RVQ module, a dual-channel decoder, and the paper presents a two-stage training procedure. Experiments show competitive performance in WER and reconstruction quality compared to existing codecs that typically excel in only one dimension at similar bitrates. The authors validate their approach through a series of ablations, arguing that reducing shared parameters between semantic and acoustic tasks is important for achieving high-quality audio reconstruction and improved text alignment.

**Strengths:**

The paper addresses the timely problem in speech LLMs of efficiently tokenizing speech while preserving both semantic content and acoustic details. As speech LLMs gain prominence, solving this problem has important implications for practical deployment. The paper attempts to validate the approach across multiple evaluation dimensions including reconstruction metrics, text alignment, cross-lingual settings, and out-of-domain scenarios.

**Weaknesses:**

The introduction would benefit from stronger motivation and clearer framing of concepts. Core concepts like "semantic tokens" and "acoustic tokens" should be defined for a broader audience, and it would help to explicitly explain why speech LLMs benefit from modeling both semantic and acoustic information simultaneously. The authors should consider establishing the problem and its importance more clearly before introducing the technical contribution. In addition, the introduction mentions that speech codecs are "the critical component" of speech LLMs, which is too narrow (e.g., see [1]). Many recent speech LLMs rely on continuous encodings and produce text, making tokenization just one of many design decisions rather than the singular focus. The manuscript would benefit from specifying the multiple paradigms in modern speech LLM research. Furthermore, the manuscript should include a compact related work section in the main body (not just the appendix) to clearly embed the contribution in prior work.

Section 2 presents a comparison of pretrained encoders to justify selecting Whisper, but has several issues. The section placement disrupts the paper's flow and would be better positioned within the method description or ablation. Additionally, the autoencoder setup (without quantization) may not reflect actual codec performance. Could the authors please justify this design choice? Also, essential details are missing, for example evaluation metrics should be defined in the main text, and dataset/training information should be summarized rather than deferred to the appendix.

Section 3 describes the XY-Tokenizer architecture and training strategy, but several claims need better support and the presentation could be clearer. The authors claim that decreasing shared parameters effectively mitigates the trade-off based on Table 2, but this conclusion is not well-supported. The three models have differences beyond just shared parameters, making it impossible to isolate this effect. Consider either conducting a controlled ablation (varying only shared parameters) or softening this claim. Also, the dual-channel encoder design needs clearer justification. Why is this architecture preferable to alternatives? Several mathematical expressions contain unclear formulations (e.g., the ASR loss describes y_t as the predicted token). Finally, critical architectural details (layer dimensions, specific module designs) are not available in the main text, making it difficult to fully understand the method. Consider summarizing key architectural choices in the main text.

The ASR probing evaluation in Section 4 is non-standard (see also questions). Also, the baseline comparisons lack context, for example were baselines trained on the same dataset? The ablation shown in Table 4 is insufficient for the core claim about shared parameters, comparing only dual-channel versus single-channel encoders rather than systematically varying parameter sharing. The evaluation metrics need stronger motivation. Why are these chosen metrics appropriate for validating your method, and how do they relate to downstream speech LLM performance?

[1] S. Arora, K.-W. Chang, C.-M. Chien, Y. Peng, H. Wu, Y. Adi, E. Dupoux, H.-Y. Lee, K. Livescu, and S. Watanabe, “On the landscape of spoken language models: A comprehensive survey,” arXiv preprint arXiv:2504.08528, 2025.

**Questions:**

Most of my questions are added in the Weaknesses section for readability. In addition, I was wondering if the authors are aware of [2] and the comprehensive set of evaluations presented in [3]? Could you justify why their custom evaluation protocol better validates their claims than adopting such a standardized evaluation setup / broader set of evaluations?

[2] Haibin Wu, Ho-Lam Chung, Yi-Cheng Lin, Yuan-Kuei Wu, Xuanjun Chen, Yu-Chi Pai, Hsiu-Hsuan Wang, Kai-Wei Chang, Alexander Liu, and Hung-yi Lee. 2024. Codec-SUPERB: An In-Depth Analysis of Sound Codec Models. In Findings of the Association for Computational Linguistics: ACL 2024, pages 10330–10348, Bangkok, Thailand. Association for Computational Linguistics.
[3] P. Mousavi, G. Maimon, A. Moumen, et al., “Discrete audio tokens: More than a survey!,” TMLR, 2025.

---

### Official Review · Reviewer_8oZU · 2025-10-30

**Soundness:** 3
**Presentation:** 3
**Contribution:** 3
**Rating:** 6
**Confidence:** 4

**Summary:**

The paper proposed XY-Tokenizer, a semantic-aware codec with reconstruction quality comparable to SOTA codecs. The core idea is to mitigate conflict between semantic and acoustic modeling by minimizing shared parameters in a multitask learning framework. To this end, the authors proposed a novel two-stage training pipeline with parallel semantic and acoustic encoder and decoder. Experiment shows that the proposed method achieves a marginally better balance between semantic and acoustic metrics compared to baseline methods.

**Strengths:**

1. XY-Codec achieves a superior balance among reconstruction quality, semantic awareness, frame rate and model size.
2. The observation that “decreasing the number of shared parameters between semantic and acoustic modeling mitigates semantic-acoustic conflict” is insightful.
3. The idea of separating semantic and acoustic channels and incorporating pretrained Whisper as the semantic encoder provides a novel solution to the semantic-acoustic conflict problem.

**Weaknesses:**

1. **Marginal Improvement**: The semantic ability and model size of XY-Tokenizer is on-par with the baseline Baichuan method, while the improvement on reconstruction quality seems marginal. There is still a significant gap between reconstruction quality of XY-Tokenizer and higher-bitrate baselines.
2. **Generalizability**: XY-Tokenizer is trained on Emilia, a speech-only dataset for TTS. Additionally, the acoustic encoder needs to be initialized with Whisper parameters. This means that XY-Tokenizer may be a speech codec rather than a general audio codec, limiting its application to semantic-rich scenarios like TTS and conversations.
3. **Inadequate Experiments**: Ablation on the post-training stage is missing. Moreover, since the improvement on reconstruction quality is not significant, it would be better if perceptual metrics like MOS could be provided.

**Questions:**

1. **Parameter Efficiency**: XY-Tokenizer utilizes a dual-encoder architecture, which raises the computational cost. This raises a question: Why don’t the LLM developers train a model with both Whisper and acoustic-only codec inputs and output only codec tokens? Granted, the token efficiency of XY-Tokenizer may be higher, but this could be mitigated by delayed stack of tokens or hierarchical transformer [1].
2. **Comparison with Similar Works**: Some recent works [2] have already proposed codecs with similar bitrates, semantic awareness and acoustic quality. The idea of dual-channel encoder [3] and utilizing Whisper encoder [4] have also been explored. What is the difference between XY-Tokenizer and these existing/concurrent works?

**References**

[1] Défossez, Alexandre, et al. "Moshi: a speech-text foundation model for real-time dialogue." *arXiv preprint arXiv:2410.00037* (2024).

[2] Yang, Dongchao, et al. "ALMTokenizer: A Low-bitrate and Semantic-rich Audio Codec Tokenizer for Audio Language Modeling." *Forty-second International Conference on Machine Learning*.

[3] Li, Jiaqi, et al. "Dualcodec: A low-frame-rate, semantically-enhanced neural audio codec for speech generation." *arXiv preprint arXiv:2505.13000* (2025).

[4] Wang, Yuanyuan, et al. "DualSpeechLM: Towards Unified Speech Understanding and Generation via Dual Speech Token Modeling with Large Language Models." *arXiv preprint arXiv:2508.08961* (2025).

---

### Official Review · Reviewer_88hE · 2025-10-31

**Soundness:** 3
**Presentation:** 3
**Contribution:** 2
**Rating:** 4
**Confidence:** 5

**Summary:**

The paper proposes XY-Tokenizer, a speech codec designed to balance semantic alignment (with text) and acoustic fidelity (audio quality) at low bitrates (~1 kbps). The model introduces:

1. A dual-tower (semantic + acoustic) encoder-decoder architecture.

2. A two-stage training pipeline: (1) multi-task pretraining combining ASR and reconstruction losses; (2) GAN-based post-training for fine-grained audio quality.

Empirical analyses comparing shared vs. separated parameters, showing that decoupling semantic and acoustic paths reduces conflicts.
Experiments on LibriSpeech and other datasets show that XY-Tokenizer achieves similar or better WER and SIM scores than baselines like SpeechTokenizer, Mimi, XCodec2.0, and BigCodec at similar bitrates.

**Strengths:**

1. The paper identifies a real problem: the semantic–acoustic tradeoff in speech codecs, which is relevant to Speech LLMs and audio foundation models.

2. The multi-stage training pipeline (ASR + reconstruction, then GAN fine-tuning) is systematically described and experimentally validated.

3. Evaluations cover semantic (ASR WER) and acoustic (SIM, STOI, PESQ) metrics. The results demonstrate improvements over baselines, particularly at low bitrate (1 kbps), suggesting good engineering effort.

**Weaknesses:**

1. Limited novelty — ASR-based loss is already standard. The central methodological component, the ASR-based semantic loss, is not new. Recent audio tokenizer such as Baichuan-Audio and CosyVoice-3 already use similar strategies — integrating an ASR-guided objective to enhance semantic representation in codecs. Thus, the contribution of XY-Tokenizer is incremental.

2. Lack of justification for using ASR supervision. The paper does not validate the necessity of the ASR-based loss.
Because the model already employs a pretrained SSL encoder (e.g., whisper, WavLM), which intrinsically captures semantically stable features, it is unclear whether additional ASR supervision provides any unique benefit. A more direct and computationally efficient alternative would be to train the decoder to reconstruct SSL features instead of ASR outputs. Without an ablation comparing ASR-based loss vs. SSL-feature reconstruction loss under the same conditions, the effectiveness of the proposed approach remains speculative.
Furthermore, introducing an LLM-based ASR head dramatically increases training cost and complexity. The paper fails to show that this added cost is justified by meaningful gains.

**Questions:**

See the weakness part.

---

### Official Review · Reviewer_SA1J · 2025-11-01

**Soundness:** 3
**Presentation:** 2
**Contribution:** 3
**Rating:** 6
**Confidence:** 4

**Summary:**

The paper presents an approach to realize low-bit rate speech codec that can preserve acoustic information while retaining the semantic information in speech. The paper is well motivated and addresses a relevant research topic. Results are compared with prior art and reference to prior work is well documented. Results based on objective metrics, demonstrate the usefulness of the proposed approach.

The technical content is clearly outlined, however it is not clear if the recipe for the tokenizer training will be shared with the research community as a repository.

**Strengths:**

The work investigates an important and relevant topic of speech tokenizer training that emphasizes not only on the content but also on the quality of the reconstructed speech. The paper clearly outlines the goals of the work presented, how it differs from prior art and compares performance with respect to prior art using relevant objective metrics.

The work addresses the weaknesses in existing tokenizer training approaches and elaborates on how it impacts either the quality or the content of the generated speech under low bit rate conditions. The proposed multi-stage multi-task approach presented in this work aims to concurrently model semantic and acoustic information.

**Weaknesses:**

The results presented in this work are convincing, however mostly focuses on objective measure. A subjective evaluation, in the form of mean opinion score would have made the content and the results presented in the paper more convincing. While objective measures are often used for assessing speech quality, but it is also true that human perceptual measures are often more reliable metrics when assessing the quality and semantic structure of generated speech.

**Questions:**

(1) Table 2 introduces the metrics used for comparing tokenizers, however SIM is not introduced before table 2, but is used in that table for comparison. Suggest, introducing the metric before its use.

(2) "..their ability to retain paralinguistic information remains under explored" >>  some of these pre-trained models have been explored successfully for paralinguistic information extraction, such as detecting emotion from speech. Suggest addressing those prior art and rephrasing this line.

(3) For the reconstruction task, only objective metrics have been used. It will be interesting to have subjective metrics such as the Mean opinion score.

---

### Official Review · Reviewer_Q3h3 · 2025-11-01

**Soundness:** 2
**Presentation:** 3
**Contribution:** 2
**Rating:** 2
**Confidence:** 5

**Summary:**

The paper introduces XY-Tokenizer, a dual-tower speech codec that combines ASR and reconstruction objectives to address the semantic-acoustic tradeoff in codec training. The approach effectively extends existing codec frameworks by adding an ASR branch on top of standard reconstruction-based training. While the motivation is clear and the formulation is coherent, the work represents a modest extension rather than a fundamentally new method. Experimental results show competitive but not superior performance to prior codecs such as SpeechTokenizer and DAC, making the practical gain and originality claims less convincing.

**Strengths:**

1. The joint modelling of ASR and reconstruction is an interesting and well-motivated approach that could inspire future research bridging reconstruction and transcription tasks.
2. The dual-tower design (semantic and acoustic encoders with minimal parameter sharing) provides a clear architectural solution to mitigate the semantic–acoustic conflict.
3. The two-stage training of ASR/reconstruction pre-training followed by GAN-based post-training offers a sound recipe to balance intelligibility and perceptual quality in speech codecs.
4. The paper's framing of the semantic-acoustic trade-off problem is relevant for speech LLM research.

**Weaknesses:**

1. The claim that “XY-Tokenizer is the first speech codec to successfully model both semantic and acoustic information effectively at low bitrates” is overstated. Prior works such as SpeechTokenizer and Mimi already combined semantic acoustic information, while XCodec model both simultaneously. The lower bitrate argument is also weakly motivated, as established codecs (e.g., SpeechTokenizer) can easily operate at lower bitrates by reducing quantizer depth.
2. In Table 3, XY-Tokenizer does not clearly outperform existing baselines. The paper instead highlights “top-2” results, where XY-Tokenizer typically ranks second. This raises doubts about whether the added complexity of the dual-tower structure and ASR decoder is justified. Moreover, using “lower bitrate” to justify weaker results is not convincing. If lower bitrate is the explanation for weaker results, the authors should include matched-bitrate experiments (e.g., XY-Tokenizer trained at 4 or 6 kbps) to demonstrate that XY-Tokenizer performs better than SpeechTokenizer/DAC under identical conditions. Without this, the claimed improvement in semantic-acoustic modelling is not substantiated.
3. XY-Tokenizer should be tested on established benchmarks such as Codec-SUPERB (full tasks), ARCH, or DASB. Relying solely on a selected LibriSpeech test-clean subset makes it difficult to assess generalization or compare to prior work.
4. The alignment claim is not well-supported. Training an auxiliary ASR branch and a separate LLM-based decoder does not necessarily guarantee better alignment between semantic tokens and acoustic reconstruction. The paper should provide clearer evidence (e.g., representational analyses) showing that the learned codes are jointly optimized for both modalities rather than independently specialized.
5. Human perceptual evaluation is missing. Objective metrics (SIM, PESQ, STOI) are not sufficient; subjective scores such as MOS, MUSHRA, or at least UTMOS should be reported to validate perceptual quality claims.
6. Missing key baselines such as EnCodec, DM-Codec, and FACodec. These are strong, contemporary codecs, and without them, it is unclear where XY-Tokenizer stands relative to the current literature.
7. Ablations and additional results appear constrained: (i) cross-lingual evaluation only on Dutch, (ii) out-of-domain testing on just 100 hours of audio, (iii) LLM-based understanding and generation tasks compared with only one or two weaker baselines.
8. Formatting issue: extensive boldface in tables makes interpretation difficult; consistent use of underlines or italics would improve readability.

**Questions:**

1. Please see the Weakness section for points where additional analysis/clarification or experimentation would strengthen the paper.
2. In the introduction, the statement “Semantic tokens, typically derived from discretized self-supervised learning (SSL) models, are considered to exhibit high alignment with text while leading to poor reconstruction” lacks a clear citation. Does any prior empirical work explicitly support this observation?

---

### Note · Authors · 2025-12-01

I have read and agree with the venue's withdrawal policy on behalf of myself and my co-authors.